# Clinical Evaluation of Non-Contrast-Enhanced Radial Quiescent-Interval Slice-Selective (QISS) Magnetic Resonance Angiography in Comparison to Contrast-Enhanced Computed Tomography Angiography for the Evaluation of Endoleaks after Abdominal Endovascular Aneurysm Repair

**DOI:** 10.3390/jcm11216551

**Published:** 2022-11-04

**Authors:** Karim Mostafa, Julian Pfarr, Patrick Langguth, Jost Philipp Schäfer, Jens Trentmann, Ioannis Koktzoglou, Robert R. Edelman, Fernando Bueno Neves, Joachim Graessner, Marcus Both, Olav Jansen, Mona Salehi Ravesh

**Affiliations:** 1Department of Radiology and Neuroradiology, University Medical Center Schleswig-Holstein, Kiel University, 24105 Kiel, Germany; 2Department of Radiology, NorthShore University Health System, Evanston, IL 60201, USA; 3Pritzker School of Medicine, University of Chicago, Chicago, IL 60637, USA; 4Feinberg School of Medicine, Northwestern University, Chicago, IL 60611, USA; 5Siemens Healthcare GmbH, 20910 Hamburg, Germany

**Keywords:** endovascular aortic aneurysm repair, non-contrast-enhanced angiography, radial quiescent-interval slice-selective (QISS) pulse sequence, contrast-enhanced computed tomographic angiography (CTA)

## Abstract

Purpose. Contrast-enhanced (CE) angiographic techniques, such as computed tomographic angiography (CE-CTA), are most commonly used for follow-up imaging after endovascular aneurysm repair. In this study, CE-CTA and non-CE QISS-MRA were compared for the first time for assessing endoleaks and aneurysms at follow-up after abdominal EVAR. Methods. Our study included 20 patients (17 male, median age 79.8 years) who underwent radial QISS-MRA and CE-CTA after EVAR at their first follow-up examination. Two interventional radiologists evaluated datasets from both techniques in each patient concerning presence of endoleaks, types of endoleaks, aneurysm diameter, and image quality. Interobserver and intermodal agreement were assessed with Cohen’s Kappa. Results. Image quality was rated as excellent or good for both modalities by both observers. Ferromagnetic embolization materials cause hyperdense artifacts in CE-CTA causing aneurysm sac diameter measurements to be inaccurate by up to 1 cm. Type 2 endoleaks with low-flow characteristics in CE-CTA were overlooked compared to radial QISS-MRA. Compared to CE-CTA, all endoleaks after abdominal EVAR were detected and classified correctly on QISS-MRA. The interobserver agreement between CE-CTA and QISS-MRA was almost perfect, except for type 2 endoleaks, where agreement was substantial. Intermodal aneurysm diameter correlate “very strongly” for both observers. Conclusions. Radial QISS-MRA is a contrast agent free technique for diagnosing and monitoring all types of endoleaks and aneurysms in patients after abdominal EVAR. It provides information about specific clinical questions concerning aneurysm diameter and presence and types of endoleaks without radiation exposure and the side effects associated with iodine-based contrast agents.

## 1. Introduction

Endovascular aortic aneurysm repair (EVAR) is a minimally invasive interventional technique for the treatment of abdominal aortic aneurysms. Using this method, a covered stent graft is inserted via delivery sheaths through the femoral arteries into the aneurysmal segment of the aorta to eliminate the pressure from the systemic circulation on the aneurysm and prevent aortic rupture. EVAR represents an excellent alternative to open surgical aortic repair and it is predominantly becoming the treatment of choice [1]. 

Although EVAR provides good long-term results, complications, such as leakage of blood into the aneurysm sac, known as endoleaks, can develop. Five types of endoleaks are known: inadequate apposition of the stent graft to the vessel wall with subsequent perfusion of the aneurysm sac in an anterograde (type Ia) or retrograde (type Ib) manner, neovascularization, or retrograde blood flow via genuine vessels into the aneurysm sac (type II), stent graft discontinuation (type III), graft material failure or defect (type IV), and an unclear increase in aneurysm sac diameter (type V) [2]. Endoleaks can promote further growth of the aneurysm sac, and, thus, result in an increased risk of aortic rupture. Therefore, it is essential to diagnose, evaluate, and treat them. Type I and II endoleaks are the most common types (Figure 1 and Figure 2). Type Ia and Ib endoleaks always require secondary intervention, while type II endoleaks only need to be treated when the aneurysm sac continues to enlarge or the patient presents with clinical symptoms [2,3]. Follow-up imaging examinations after EVAR needs be performed within 30 days after the procedure to stratify the patient to a low-, intermediate-, or high-risk group [3,4]. Depending on the risk stratification, different follow-up algorithms are applied [3]).

There are various invasive, minimally-invasive, and non-invasive imaging techniques available for follow-up after EVAR. Digital subtraction angiography (DSA) is an invasive procedure, which cannot provide adequate information about the diameter of the aneurysm sac [5]. Nowadays, contrast enhanced computed tomographic angiography (CE-CTA) represents a standard clinical technique for follow-up imaging and is recommended immediately after, as well as five years after EVAR [3,5]. CE-CTA is widely available, but it is not suitable as the method of choice for long-term follow-up imaging due to the exposure to ionizing radiation and the side effects of iodine-based contrast agents [3,4]. CE-magnetic resonance angiography (MRA) and non-CE or CE-color-doppler ultrasound (CDUS) constitute radiation free alternatives to CE-CTA. 

The US-based techniques have some technical limitations, however, such as small field of view, strong operator dependence, low penetration depth, and susceptibility to artifacts [3,6]. In 2019, Salehi Ravesh et al. investigated the use of non-CE radial quiescent-interval slice-selective (QISS) MRA for evaluating aortic aneurysms and their endoleaks [7]. This MRA technique was published in 2017 for the first time by Edelman et al. for imaging pulmonary embolisms [8]. In the present prospective study, we compared radial QISS-MRA versus CE-CTA in detecting and grading endoleaks and determining aneurysm diameters in patients after abdominal EVAR.

## 2. Material and Methods

The local Institutional Review board approved this prospective single-center study in September of 2018 (No. D576/18). All study participants gave informed consent in a written form in accordance with the ethical standards laid down in the 1964 declaration of Helsinki and its later amendments.

### 2.1. Study Design and Study Population

This single-center prospective study conducted between 2018 and 2020 included 20 patients (17 male, median age of 79.8 years) for which a non-fenestrated aorto-biiliac prosthesis was implanted at our medical center. These patients were then also examined in the framework of follow-ups for abdominal EVAR one month after implantation with non-CE radial QISS-MRA, in addition to CE-CTA. Patients in unstable cardiopulmonary conditions with a cardiac pacemaker, large-sized ferromagnetic materials in the thoracic region or further contra-indications for MRI were excluded from our study. Aneurysm diameters and presence of endoleaks were assessed and endoleak types were graded. In addition, the quality of the images was evaluated for each modality.

### 2.2. CE-CTA

CE-CTA images were acquired using a dual-source 128-row CT system. Images were acquired with a tube voltage of 100 kV, a tube current time product of 59 mAS and a slice collimation of 2.0 mm × 128 mm × 0.6 mm with a spiral pitch factor of 1.2. Here, 1 mL/kg body weight iodine-based contrast agent (Imeron 350, Bracco Imaging, Milano, Italy) was injected intravenously at an injection rate of 4 mL/s. The images were acquired in an arterial and a portal-venous phase.

### 2.3. Non-CE Radial Electrocardiogram Triggered QISS-MRA

QISS-MRA images were acquired on a 1.5 Tesla MRI system (Siemens Healthineers, Erlangen, Germany) in axial, sagittal, and coronal planes under breath hold using an 18-channel torso coil. The imaging protocol parameters are listed in Table 1.

Compared to conventional MRA, QISS-MRA is a 2D cardiac-triggered non-CE MRA technique that suppresses stationary tissue and venous blood flow, making it possible to visualize any arterial inflow using a balanced steady-state precision readout [8].

### 2.4. Image Analysis

Two board-certified interventional radiologists (J.P.S. and J.T.), each with more than 10 years of clinical experience, systematically assessed the CE-CTA and QISS-MRA images. A total of 40 datasets (20 per angiographic technique) were evaluated by both observers. A time frame of 4 weeks was kept between assessments of the datasets of each imaging modality to ensure that previously made observations did not interfere with the latter ones. Aneurysm diameters were measured in transverse images from outer wall to outer wall. Image quality was graded into 4 classes (Figure 3).

Grade 1: Poor arterial signal and poor vascular contrast with undefinable outlining of aneurysm. Non-diagnostic.Grade 2: Ill-defined vessel-borders with suboptimal image quality for diagnosis.Grade 3: Minor inhomogeneities, and the vessel outlining is clearly visible.Grade 4: Excellent image without artifacts.

### 2.5. Modality-Specific Diagnosis of Endoleaks

In the first step, numbers and types of endoleaks diagnosed by each observer were individually compared for QISS-MRA and CE-CTA, respectively. Subsequently, the interobserver agreement of findings per imaging modality was assessed.

### 2.6. Observer-Specific Intermodal Assessment of Endoleak and Endoleak Subtype Diagnosis

In the second step, findings of each observer from QISS-MRA were compared to their corresponding findings from CE-CTA, in terms of the presence of endoleaks and determination of endoleak type.

### 2.7. Observer-Specific Intermodal Assessment of Aneurysm Diameter

In a third step, maximum diameters of aneurysms were determined in transverse datasets from outer wall to outer wall according to both angiographic techniques. The aneurysm diameters in CE-CTA and radial QISS-MRA were compared for each observer.

### 2.8. Statistical Analysis

The observations were categorized into imaging modality, observers, and endoleak types. Normality (Gaussian distribution) was tested for all variables using the Shapiro–Wilk test. Normal distribution was proven for the aneurysm diameters in CE-CTA and radial QISS-MRA for both observers. The interobserver and intermodality agreement between CE-CTA and radial QISS-MRA concerning the detection of endoleaks and their subtypes were assessed using Cohens’ Kappa (κ). Kappa values ≤ 0 indicate no agreement, 0.01–0.20 none to slight, 0.21–0.40 fair, 0.41–0.60 moderate, 0.61–0.80 substantial, and 0.81–1.00 almost perfect agreement [9]. Correlation of aneurysm size was assessed using the Spearman’s correlation. A ρ-value > 0.8 was considered a very strong correlation. Significance was defined at a *p*-value < 0.05.

## 3. Results

### 3.1. Image Quality Assessment

For CE-CTA, observer 1 graded the image quality in 90.5% (19/21) as “excellent (grade 4)” and in 9.5% (2/21) as “good (grade 3)”, while observer 2 graded 100% (21/21) as “excellent (grade 4)”. For QISS-MRA, observer 1 graded the image quality in 85.7% (18/21) as “excellent (grade 4)” and in 14.3% (3/21) as “good (grade 3)”, while observer 2 graded 76.2% (16/21) as “excellent (grade 4)”, 19.0% (4/21) as “good (grade 3)” and 4.8% (1/21) as “fair (grade 2)”. No images were graded as “nondiagnostic (grade 1)” for either modality.

### 3.2. Comparison between the Imaging Modalities concerning Aneurysm Size

The measured mean diameter of aneurysm by observer 1 was 5.8 cm (range 3.7–10.0 cm) in CE-CTA and 5.8 cm (range 3.8–9.8 cm) in QISS-MRA. Statistically, the agreement between the CE-CTA and QISS-MRA results was “very strong” (ρ = 0.96, *p* < 0.01). The assessed mean diameter of aneurysm by observer 2 was 5.7 cm (range 3.5–10.0 cm) in CE-CTA and 5.7 cm (3.8–9.0 cm) in QISS-MRA. The correlation between these two findings was also “very strong” (ρ = 0.92, *p* < 0.01).

In one patient, a difference of up to 1 cm in aneurysm size was determined between modalities and observers due to metal artifacts in CE-CTA (Figure 4).

### 3.3. Number of Diagnosed Endoleaks and Endoleak Types

Observer 1 found a total number of 12 endoleaks in CE-CTA (type Ia: 1, type Ib: 3, type II: 8) and 16 in QISS-MRA (type Ia: 1, type Ib: 3, type II: 11, and type V: 1).

Observer 2 detected a total number of 11 endoleaks in CE-CTA (type Ia: 1, type Ib: 3, and type II: 7) and 14 endoleaks in QISS-MRA (type Ia: 1, type Ib: 3, type II: 9, and type V: 1).

### 3.4. Observer-Specific Intermodal Assessment of Endoleaks and Endoleak-Subtype Diagnosis

The agreement in the diagnosis of endoleaks among the imaging modalities for each observer (Table 2) was statistically almost perfect for all endoleak types. For type 2, endoleaks the agreement was “substantial”, since both observers found more type 2 endoleaks in QISS-MRA than in CE-CTA (Observer 1 κ = 0.71, *p* < 0.01; Observer 2 κ = 0.79, *p* < 0.01). In two patients, both observers diagnosed a type 2 endoleak only in QISS-MRA (Figure 2). This observation was subsequently considered by consensus between the two observers to be the correct diagnosis, as opposed to a false positive finding.

### 3.5. Modality-Specific Agreement on Endoleak Diagnosis

The interobserver agreement for diagnosis of all detectable endoleak types was perfect overall for both modalities except for type II endoleaks. For type II endoleaks, the interobserver agreement was significantly “substantial” for QISS-MRA and significantly “almost perfect” for CE-CTA. The results are listed in Table 3.

## 4. Discussion

Contrast agent and radiation free vessel imaging techniques are an important development of the past decade. In 2010, cartesian QISS-MRA, and its later variants, have been shown to provide reliable diagnostic results in various vascular territories [5,7,8,10]. In 2019, Salehi Ravesh et al. demonstrated that 2D radial QISS-MRA is a promising non-CE technique for post-interventional monitoring of patients with an endovascular aortoiliac prosthesis [7]. In this present study, radial QISS-MRA was evaluated for the first time and compared to CE-CTA for diagnosing endoleaks and determining aneurysm diameter in patients after abdominal EVAR.

The main findings of this study are the following: (1) In comparison with CE-CTA, all endoleaks after abdominal EVAR can be detected and classified as type 1 to type 5 using non-CE radial QISS-MRA. (2) Some type II endoleaks could only be detected by QISS-MRA, which was most likely due to the low-flow characteristics of the endoleaks with an influx of less than 0.25 mL/s [11]. (3) Ferromagnetic embolization materials that cause local signal dropouts in QISS-MRA images do not diagnostically affect the accuracy of determining the aneurysm sac diameter compared to CE-CTA. (4) In patients stratified as intermediate-risk after EVAR, QISS-MRA can be used as an alternative imaging method for monitoring the aneurysm diameter and diagnosing new endoleaks alongside duplex ultrasound, especially in cases where ultrasound shows inconclusive results or aneurysm growth is suspected. In patients stratified as low-risk, QISS-MRA represents a contrast- and radiation-free alternative to CE-CTA at the five-year follow-up examination.

### 4.1. Aneurysm Diameter Measurement

Aneurysm diameters determined by both observers in CE-CTA and QISS-MRA were strongly and significantly correlated. Prior research has shown large variability in the reporting of the aneurysm diameter, particularly in studies that included diameter measurements based on outer wall to outer wall, inner wall to inner wall, and anterior outer wall to posterior inner wall [3,12]. It has been recommended that the maximum aneurysm diameter is to be measured through the centerline in reconstructed images perpendicular to the blood flow [3,6,12]. In accordance with the literature, we determined the maximum transverse diameters of aneurysm sacs measured from outer wall to outer wall through the vessel centerline in both QISS-MRA and CE-CTA.

Large hyperdense metallic artifacts can occur in CE-CTA, owning to ferromagnetic embolization materials, for example, and can cause inaccurate determination of the aneurysm sac (Figure 4). Moreover, endoleaks near these artifacts will be overlooked. In QISS-MRA, as in other MRI techniques, the ferromagnetic embolization material produces locally reduced signal intensity, which is related to susceptibility artifacts. Despite this local signal dropout, the aneurysm sac was more accurately delineated in QISS-MRA than in CE-CTA, and type II endoleaks could clearly be detected. Based on our results, the presence of ferromagnetic embolization materials in an aneurysm sac can cause interobserver disagreement concerning the diameter of the aneurysm in CE-CTA by more than 1 cm, while the corresponding measurements in QISS-MRA were similar.

### 4.2. Modality-Specific- and Observer-Specific Agreement on Endoleak Diagnosis and Image Quality

Our results show that the interobserver agreement concerning detection and grading of endoleaks for CE-CTA and QISS-MRA was almost perfect. Koike et al. demonstrated almost perfect interobserver agreement on endoleak grading in CE-CTA in 8 patients after abdominal EVAR with type I, II and III endoleaks, which is in good accordance with our findings [13].

However, the intermodal agreement between QISS-MRA and CE-CTA in the diagnosis of type II endoleaks was moderate in our study. In two patients, both observers diagnosed a most likely low-flow type II endoleak only in radial QISS-MRA, but not in CE-CTA (Figure 2). These type II endoleaks in QISS-MRA were subsequently considered by consensus to be the correct diagnosis. Owing to retrograde blood flow and associated low blood pressure conditions, some type II endoleaks appear to have a very low blood inflow velocity, making them difficult to detect on CE-CTA [5]. The sensitivity of CE-CTA decreases when contrast inflow is less than 0.25 mL/s [11]. As displayed in Figure 2, QISS-MRA does not have this limitation, as any freely moving liquid can be imaged.

### 4.3. Visualization of Endoleaks and Image Quality

The detection of endoleak types I–V can be achieved using CE-CTA [12]. In addition, CE-CTA is a useful tool in diagnosing stent-graft defects. The disadvantages of CE-CTA are well known, namely radiation and contrast agent exposure of the patient, diagnostic disadvantages in the presence of artifacts, as well as the inability to determine the flow direction of endoleaks [12]. Based on our research, it seems that some type II endoleaks with low blood inflow velocity are difficult to see on CE-CTA, most likely due to the aforementioned decreased sensitivity of CE-CTA when the influx of contrast media is lower than 0.25 mL/s [10,11]. Theoretically, acquisition of CT images in a “very late phase” with 160 or 240 s after contrast administration, for example, may be a method to overcome this limitation. However, this could only be possible at the cost of further exposure to radiation. Furthermore, it is unclear how long the contrast media will remain inside the aneurysm sac, since it may also be subjected to the flow phenomena. Although both observers rated the overall quality of QISS-MRA images slightly lower than for CE-CTA (Figure 3), the relevant clinical questions regarding the size of the aneurysm and presence of endoleaks could always be answered.

### 4.4. Avoidance of Radiation and Iodine-Based Contrast Agents

The clinically relevant advantage of QISS-MRA compared to CE-CTA, however, lies in the fact that the patient is neither being exposed to radiation nor to an iodine-based contrast agent. Although CE-CTA is an important clinical imaging technique with clear indications in the setting of patients after abdominal EVAR, repeated exposure to radiation or contrast agent is not negligibly harmless [3,14]. In an effort to reduce contrast agent exposure, Bobadilla et al. showed in their cohort that non-CE CT was sufficient to monitor aneurysm diameters after EVAR over the course of follow-up [15]. In contrast, QISS-MRA does not represent a cause for concern and can be repeated as often as clinically indicated.

In accordance with the European Society of Vascular Surgery guidelines, a QISS-MRA examination presents a useful tool for follow-up imaging of intermediate- and low-risk patients after abdominal EVAR. In case a type 2 endoleak is diagnosed in the immediate postinterventional imaging and the patient is subsequently stratified as intermediate-risk, QISS-MRA can be used rationally to monitor the endoleaks, as well as the aneurysm diameters annually, alongside duplex ultrasound, to identify patients with aneurysm or endoleak progress and establish an indication for treatment. This is especially helpful in cases with inconclusive ultrasound findings due to its known limitations [3]. Additionally, in all patient groups, QISS-MRA is a radiation and contrast free alternative to CE-CTA at the five-year follow-up appointment, especially in the setting of renal insufficiency or known allergy to contrast media. When applying this concept, radiation exposure and contrast media application can be avoided, while objective and reliable diagnostic imaging is provided.

### 4.5. Clinically Efficient Measurement Time

The use of QISS-MRA for a follow-up imaging of patients after abdominal EVAR can save time for both patients and clinical staff by eliminating the need to administer contrast agents and medically monitor the patients during and after examination. Moreover, patients can be directly referred to the hospital or imaging center where QISS-MRA is performed, as there is no need for a pre-imaging invasive blood test to determine thyroid and kidney function. There are no risks of allergic reactions to contrast agent or impaired renal function. A venous vascular access, needed for CE-CTA, can be avoided. The total measurement time for the multi-slice three-plane QISS-MRA examination used here is about 10 minutes. Measurement time however can be shortened in the future by using a highly undersampled radial k-space trajectory [16], or by acquiring only one anatomical view (e.g., transverse) and reconstructing the other two views from this measured view. As reported in the Section 3, this idea is not diagnostically suitable for 2D QISS-MRA, because the other two reconstructed planes from the transverse 2D datasets will show horizontal line artifacts (Figure 5). These artifacts may impair detection and grading of endoleaks and affect the measurement of aneurysms. This technical problem can be solved by using a 3D QISS-MRA acquisition.

### 4.6. Study Limitations

The number of our study patients and the number of patients with different endoleak types was small, and based on the single-center design of this study.

## 5. Conclusions

Radial QISS-MRA can provide information about specific clinical questions concerning aneurysm diameters and the presence of all types of endoleaks, without radiation exposure or side effects associated with iodine-based contrast agents. This contrast and radiation free technique offers an alternative to CE-CTA for reliably diagnosing and monitoring aneurysms and their endoleaks in patients after abdominal EVAR. It can also be implemented in existing follow-up algorithms at various time points.

## Figures and Tables

**Figure 1 jcm-11-06551-f001:**
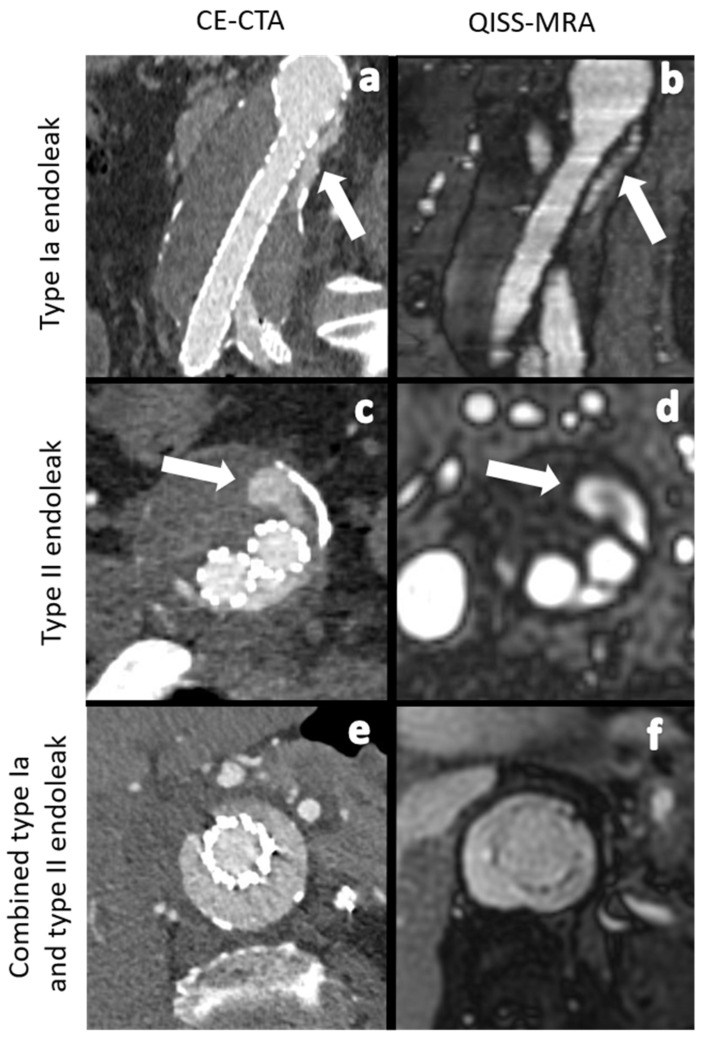
Comparison between CE-CTA (**left**) and 2D non-CE radial QISS-MRA (**right**) concerning the detection of various endoleak types. The white arrows indicate a type Ia endoleak in images (**a**,**b**) and a type II endoleak in images (**c**,**d**). In images (**e**,**f**) CT- and QISS-MRA studies of a patient with a combined type Ia and type II endoleak can be seen.

**Figure 2 jcm-11-06551-f002:**
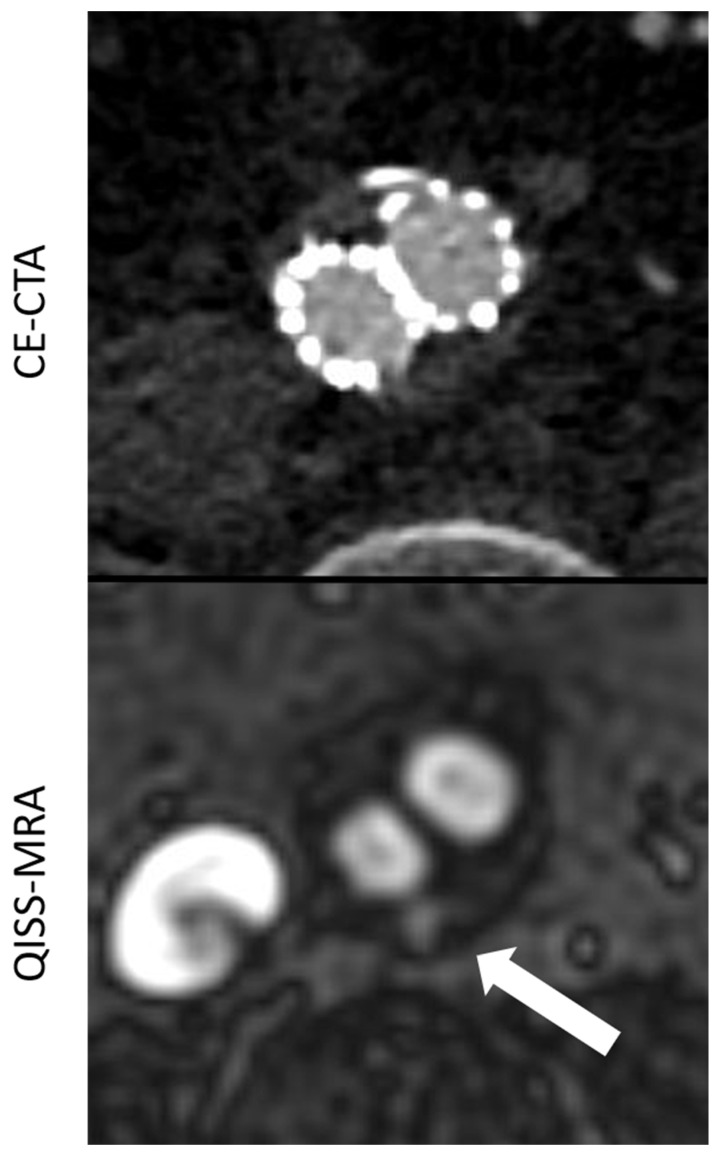
Comparison between CE-CTA and 2D non-CE radial QISS-MRA concerning the visualization of a type II endoleak. This endoleak could only be detected in QISS-MRA—which is most likely due to low-flow characteristics of the endoleak (white arrow).

**Figure 3 jcm-11-06551-f003:**
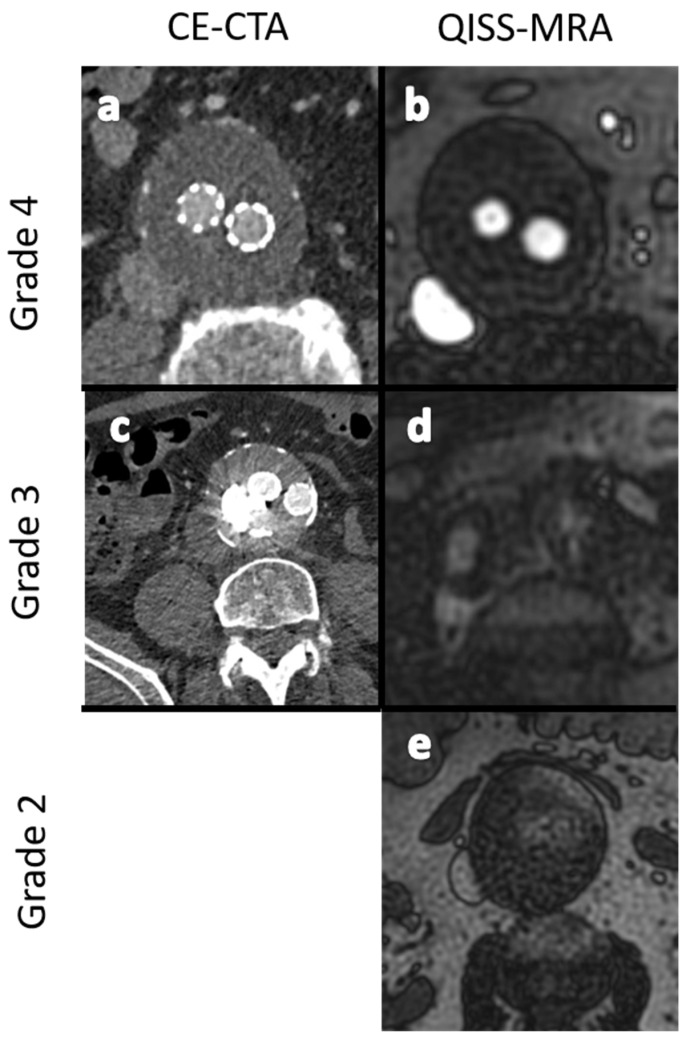
Exemplary images for the grading of image quality in grades 2 to 4. (**a**,**b**): -Excellent image quality (grade 4). (**c**,**d**): Minor inhomogeneities affecting the outlining of the vessels (grade 3). (**e**): Ill-defined vessel-borders with suboptimal image quality for diagnosis (grade 2). No dataset showed nondiagnostic image quality (grade 1).

**Figure 4 jcm-11-06551-f004:**
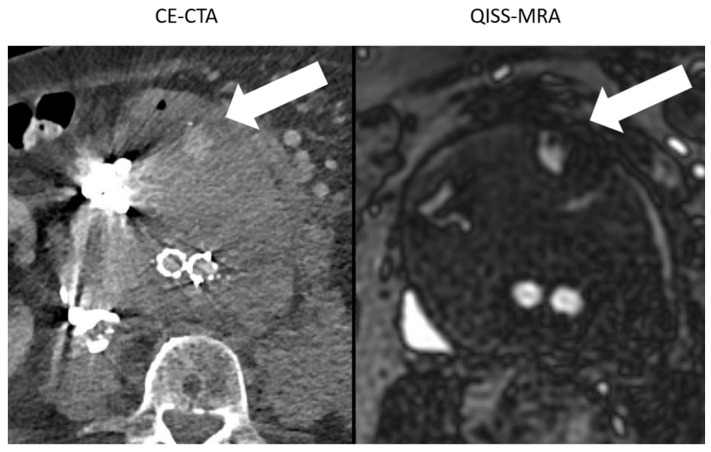
Comparison between the image quality of CE-CTA and 2D non-CE radial QISS-MRA in the presence of metallic substances. In CE-CTA, metal artifacts caused by embolization material impair the outlining of the aneurysm sac. Therefore, the adjacent type II endoleak on the hyperdense metallic artifacts can be overlooked (white arrow). In contrast, both aneurysm sac and endoleak were clearly visualized using QISS-MRA (white arrow).

**Figure 5 jcm-11-06551-f005:**
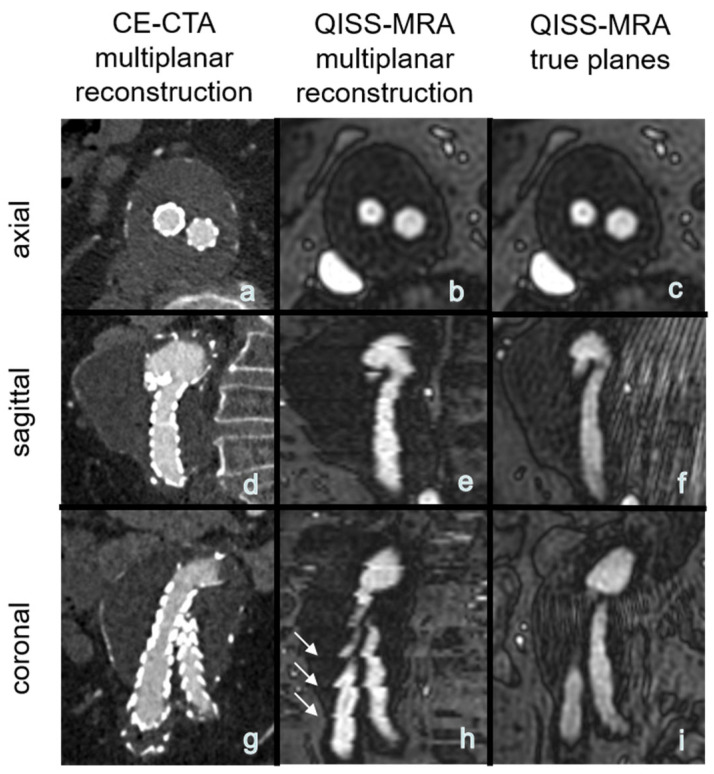
(**a**–**h**): Exemplary multiplanar CE-CTA and 2D non-CE radial QISS-MRA-images of an aneurysm. Note that the horizontal hyperintense lines are due to missing data for a 3D reconstruction, impairing aneurysm size measurements in coronal and sagittal QISS-MRA images (white arrows, image (**h**)) compared to the true (measured) sagittal and coronal planes (**f**,**i**).

**Table 1 jcm-11-06551-t001:** MR protocol parameters used in this study. The repetition time is abbreviated to TR, echo time to TE, band width to BW, flip angle to FA, and generalized auto calibrating partially parallel acquisition to GRAPPA.

Protocol Parameters	
**TR/TE (ms)**	993.9/1.7
**Acquisition matrix**	318 × 318
**Reconstructed voxel size (mm³)**	1.1 × 1.1 × 2.5
**BW (Hz/Px)**	1359
**Slice orientation**	Transverse, sagittal, coronal
**Distance factor (%)**	−20
**FA (°)**	120
**GRAPPA acceleration factor/reference line**	---
**Other**	radial balanced steady-state free precession (bSSFP) readout views: 200 chemical shift-selective fat suppression Thickness of in-plane inversion using a frequency offset corrected inversion (FOCI) Radio-Frequency (RF) pulse: 3.75 mms time from in-plane and venous saturation to acquisition of central k-space (TI): 650 ms trigger delay (TD): 0 ms bSSFP repetition time: 3.4 ms
**total acquisition time (min:sec) depending on the heart rate**	1:00

**Table 2 jcm-11-06551-t002:** Observer-specific intermodal assessment.

Observer 1					Observer 2				
	QISS-MRA	CE-CTA	κ-Value	*p*-Value		QISS-MRA	CE-CTA	κ-Value	*p*-Value
Type Ia	1 (6.2%)	1 (8.3%)	1		Type Ia	1 (7.1%)	1 (9.0%)	1	
Type Ib	3 (18.7%)	3 (25.0%)	1		Type Ib	3 (21.4%)	3 (27.0%)	1	
Type II	11 (68.7%)	8 (66.7%)	0.71	<0.01	Type II	9 (64.2%)	7 (63.6%)	0.79	<0.01
Type III	0 (0%)	0 (0%)			Type III	0 (0%)	0 (0%)	1	
Type V	1 (6.2%)	0 (0%)			Type V	1 (7.1%)	0 (0%)		
Total number	16	12			Total number	14	11		

Data are presented as number (*n*) and percentage (%). The κ- and *p*- values refer to Cohens’ Kappa.

**Table 3 jcm-11-06551-t003:** Modality-specific interobserver assessment.

QISS-MRA					CE-CTA				
	Observer 1	Observer 2	κ-Value	*p*-Value		Observer 1	Observer 2	κ-Value	*p*-Value
Type Ia	1 (6.2%)	1 (7.1%)	1		Type Ia	1 (8.3%)	1 (9.0%)	1	
Type Ib	3 (18.7%)	3 (21.4%)	1		Type Ib	3 (25.0%)	3 (27.2%)	1	
Type II	11 (68.7%)	9 (64.2%)	0.80	<0.01	Type II	8 (66.7%)	7 (63.6%)	0.89	<0.01
Type III	0 (0%)	0 (0%)	1		Type III	0 (0%)	0 (0%)	1	
Type V	1 (6.2%)	1 (7.1%)	1		Type V	0 (0%)	0 (0%)	1	
Total number	16	14			Total number	12	11		

Data are presented as number (*n*) and percentage (%). The κ- and *p*- values refer to Cohens’ Kappa.

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
