# Peer review of "Clinical Evaluation of Non-Contrast-Enhanced Radial Quiescent-Interval Slice-Selective (QISS) Magnetic Resonance Angiography in Comparison to Contrast-Enhanced Computed Tomography Angiography for the Evaluation of Endoleaks after Abdominal Endovascular Aneurysm Repair"

_jcm, 2022, doi:10.3390/jcm11216551_

Round 1
Reviewer 1 Report
To editors and reviewers
Clinical evaluation of non-contrast-enhanced QISS magnetic resonance angiography in comparison to contrast-enhanced computed tomography angiography for the evaluation of endoleaks after abdominal endovascular aneurysm repair.
- This is a very interesting manuscript that can be considered for publication in Journal of Clinical Medicine. The manuscript is appropriate with aims and scope of journal.
- I suggested some revisions below and after revisions the manuscript can be published.
1) Authors checked and revised all references and citations in your manuscript as MDPI guideline.
2) Add IRB date and Helsinki Declaration.
3) Due to small population, conclusion should be soften. Please revise.
Sincerely
Author Response
Dear Reviewer 1,
Thank you very much for reviewing our work and even more for your positive feedback! Please read my detailed responses to your comments below.
1) We checked all references and citations and revised them to fit the MDPI guidelines.
2) The date of the IRB decision was added at the beginning of the "Material and Methods" section. A statement declaring our adherence to the Helsinki Declaration can be found in the following sentence in the "Material and Methods" section.
3) Thank you for your comment. We tried to formulate the conclusion section to your liking, however our research shows that QISS-MRI can diagnose all types of endoleaks after abdominal EVAR with a statistically significant correlation to CE-CTA, which is considered as gold standard technique. This finding of our study would not change if a higher number of patients was included in the study, since we could already show this effect in our population.
Thank you again for reviewing our work.
Kind regards,
Karim Mostafa
Reviewer 2 Report
A pretty good paper about endoleaks after EVAR, correlated with MRA vs CTA.
CE-CTA and non-CE MRA were compared for assessing endoleaks and aneurysms after abdominal EVAR,relevant original and interesting topic, this identity was first time researched and a new modality.
Paper is well written ,however needs a minör english editing.Text is clear and easy to read, conclusions are consistent with the presented discussions and arguments, posing the answers of main question those were addressed in the manuscript.
Author Response
Dear Reviewer 2,
Thank you very much for reviewing our work and for your positive feedback. Prior to submission we had the manuscript spell-checked by a native english speaker, however we made sure to check it one more time to correct any errors.
Once again, thank you very much for the positive feedback.
Kind regards,
Karim Mostafa
Reviewer 3 Report
QISS-MRA is a cardiac-gated, non-contrast inflow technique especially designed for peripheral MRA. The authors investigated the utility of QISS-MRA in the postoperative evaluation of EVAR in comparison to CE-CT in 20 patients. As QISS-MRA has better detection of slowly flowing blood, good-visualization were expected.
Major comment
There have not been many reports of similar studies, and I would consider this paper to be worthy enough to be reported. The result that QISS-MRA was able to detect more type 2 endoleak seems reasonable. On the other hand, the difference of 2 cases (11 vs. 9 out of 20) in the number of type 2 endoleak diagnosed between the observers seems excessive to say "almost perfect," even though the results are considered statistically consistent (table 2).
Minor comment
The order of appearance in Table 3 and Table 2 should be changed.
The percentages in Table 3 are missing in several places.
Author Response
Dear Reviewer 3,
Thank you very much for reviewing our work and for your comments. Please find the point-by-point responses down below.
Major comment:
Thank you very much for this comment. After reviewing this paragraph, we edited it for clarification.
For QISS-MRA, the interobserver agreement for type 2 endoleak diagnosis was significantly "substantial" at a kappa-value of 0.80.
For CE-CTA, the interobserver agreement for type 2 endoleak diagnosis was significantly "almost perfect" at a kappa-value of 0.89.
We chose the definitions of the degree of agreement according to kappa-values in agreement to reference number 9 (McHugh ML. Interrater reliability: the kappa statistic. Biochem Med (Zagreb). 2012;22(3):276-282.)
We hope that this explanation could clarify this matter and thank you again for the comment.
Minor comment:
As per your suggestions, the order of appearance of table 2 and 3 was changed and the missing percentages were added.
Once again, thank you for your comments and we hope that the changes made are to your liking.
Kind regards,
Karim Mostafa